# Comparative Transcriptomics Provides Insight into the Neuroendocrine Regulation of Spawning in the Black-Lip Rock Oyster (*Saccostrea echinata*)

**DOI:** 10.3390/ijms262010032

**Published:** 2025-10-15

**Authors:** Md Abu Zafar, Saowaros Suwansa-ard, Aiden Mellor, Max Wingfield, Karl Reiher, Abigail Elizur, Scott F. Cummins

**Affiliations:** 1Centre for Bioinnovation, University of the Sunshine Coast, Maroochydore, QLD 4558, Australia; maz004@student.usc.edu.au (M.A.Z.); ssuwansa@usc.edu.au (S.S.-a.); aelizur@usc.edu.au (A.E.); 2School of Science, Technology and Engineering, University of the Sunshine Coast, Maroochydore, QLD 4558, Australia; 3Department of Aquaculture, Faculty of Fisheries, Hajee Mohammad Danesh Science and Technology University, Dinajpur 5200, Bangladesh; 4Department of Primary Industries, Bribie Island Research Centre, 144 North Street, Woorim, QLD 4507, Australia; aiden.mellor@dpi.qld.gov.au (A.M.); max.wingfield@dpi.qld.gov.au (M.W.); karl.reiher@dpi.qld.gov.au (K.R.)

**Keywords:** black-lip rock oyster, reproduction, spawning, transcriptome, gene expression

## Abstract

The black-lip rock oyster, *Saccostrea echinata*, is an emerging aquaculture species; however, difficulties in regulating their gonad conditioning to full maturation and spawning have impacted industry progress. Addressing this challenge requires a deeper understanding of the molecular mechanisms underlying reproduction, particularly the signalling molecules (e.g., neuroendocrine hormones) that regulate gonad development and spawning, which remains poorly understood in this species. Therefore, we investigated the molecular neuroendocrine regulation of gonad maturation in *S. echinata* through the analysis of gonad histological changes correlated with gene expression in the visceral ganglia and gonad (of male and female) at pre- and post-spawn stages. Our targeted analysis of neuropeptide genes demonstrated that only *LASGLVamide* showed significant differential expression, being upregulated in the pre-spawn female gonad. Of the 26 reproductive-related genes identified, four were significantly upregulated in female gonad (*SOX9*, *Dax1*, *Nanos-like*, and *Piwi-like*), while an *insulin-like peptide receptor* was elevated in male visceral ganglia at post-spawn. Untargeted investigation revealed numerous transmembrane receptors significantly upregulated in the pre-spawn ovary, such as receptors for thyrotropin-releasing hormone, metabotropic glutamate, and 5-hydroxytryptamine, while mesotocin and oxytocin receptors were upregulated in pre-spawn male gonads. At the post-spawn stage, the visceral ganglia displayed upregulation of genes encoding stress-related proteins such as superoxidase dismutase and DnaJ homologue subfamily A member 1. These findings provide important insights into the complexities of neuroendocrine signalling molecules and establish a molecular foundation to guide selective breeding and broodstock management strategies that will support sustainable aquaculture development of black-lip rock oyster.

## 1. Introduction

Oysters are considered an integral part of the aquaculture industry worldwide, with a trade value of US 125 million in 2023 [1]. In Australia, oyster production is dominated by the cultivation of two species, the native Sydney rock oyster (*Saccostrea glomerata*) and the introduced Pacific oyster (*Crassostrea gigas*) [2]. However, mass mortality events caused by disease prevalence are severely impacting the production of these species. To address these issues, the NSW oyster industry is actively advancing through selective breeding particularly for traits such as disease resistance as well as faster growth rates for both species [3,4,5,6]. Therefore, it is highly recommended to consider the aquaculture potential of other indigenous species such as black-lip rock oyster (*Saccostrea echinata*) that will significantly facilitate the industry’s expansion into new coastal areas, including the tropics [7,8].

The *Saccostrea echinata* is a prominent aquaculture species due to its delicious taste and fast growth rate, which has recently drawn researchers’ attention as a new aquaculture commodity in Australia [9]. This species can be found throughout a range of locations extending from Japan to New Caledonia, across northern Australia, from Cone Bay in Western Australia to Bowen in Queensland [10]. Aquaculture of *S. echinata* has largely relied on wild catch juvenile spat [11], primarily due to poor larval survivability [12,13] and a low settlement rate [14]. While the establishment of hatchery protocols has immensely improved *S. echinata* aquaculture [15], commercial hatchery production of this species, where spawning induction and larval settlement are challenging compared to the other commercial oyster species, is yet to be improved.

To date, *S. echinata* research has been devoted to understanding their population genetics [10,11], reproductive cycle in wild populations [16], larval development [13,14], and rearing conditions to optimise larval growth and survival [17,18]. In addition, multiple spawning induction methods were assessed, including physical manipulations (emersion, increased temperature and reduced salinity), chemical induction (addition of sperm, sperm extract, serotonin and neuropeptides), a combination of physical and chemical induction (reduced salinity and addition of sperm), and strip spawning [19]. However, there are still difficulties in the spawning induction of *S. echinata*, which impacts the ability to generate juveniles at a commercial level. This could be overcome by increasing our understanding of the neuroendocrine regulation of *S. echinata* reproduction, particularly in a spawning event. For example, the identification of key signalling molecules, such as reproductive neuropeptides. These compounds are secreted from the neural ganglia into the haemolymph via a network of neurohemal organs, upon which they are distributed to target organs, including the gonad [20]. In oysters, the presence of neuropeptides has been well-documented [21], including those that regulate key reproductive processes such as gonad maturation and spawning [22], which is regulated by the neuroendocrine system. However, our understanding of the neuroendocrine mechanisms underlying gonad maturation and spawning in *S. echinata* is limited, and addressing this gap is vital for advancing broodstock management and improving spawning success in this emerging aquaculture species.

Neuroendocrine processing in oysters is driven by a network of ganglia, primarily consisting of the cerebral, visceral, and pedal ganglia [23]. The visceral ganglia (VG), located in close association with the posterior adductor muscle, are responsible for controlling the visceral mass, including the gonad [24]. Significant regulatory processes are thought to occur between the VG and gonad prior to and following gamete (sperm and eggs) spawning, which occurs when external environmental and endogenous conditions are appropriate. Gonads at the ripe stage (pre-spawn) refers to the period leading up to gonad maturation where gametes are fully mature and awaiting release via an endogenous trigger. Immediately after the release of gametes, but before the recovery stage, the gonad condition is considered to be at a partial spent (post-spawn) stage.

In this study, we aimed to better understand the neuroendocrine regulation of spawning in *S. echinata*. To obtain this, we established a reference transcriptome derived from VG and gonads, which was then used in the targeted identification of oyster neuropeptide and reproduction-associated genes. In addition, differential gene expression at pre- and post-spawn stages provided insights into molecular mechanisms that may regulate spawning, with implications in fundamental research and applied aquaculture practices.

## 2. Results

### 2.1. Gonad Histology at Pre- and Post-Spawn S. echinata

To assess the cellular changes between pre- and post-spawn gonads, histological analysis was performed (Figure 1). In the pre-spawn stage males (Figure 1A), testes largely consisted of spermatozoa that were arranged radially with tails toward the lumen, creating a swirling pattern. Spermatozoa occupied most of the spermatogenic follicle, and acidophilic contents within the lumen were prominent. In females at pre-spawn stage (Figure 1B), ovaries were replete with mature oocytes, with each characterised by a discrete nucleus and polygonal shape, featuring distinct nuclei in pre-spawn stages. Conversely, post-spawn testes and ovaries (Figure 1C and Figure 1D, respectively) were occupied by collapsed follicles, whereas interstitial connective tissue became prominent. In post-spawn testes (Figure 1C), the typical radial arrangement of spermatozoa was lost, and only a few spermatozoa and acidophilic contents within the spermatogenic follicles were characterised. Post-spawn ovaries exhibit loosely organised gametes within the follicles (Figure 1D). Free pre-spawn oocytes are prominent in these follicles.

### 2.2. Overview of Reference Transcriptome Assembly

Total RNA was isolated from female and male *S. echinata* gonads and VG at pre- and post-spawn stages (*n* = 3 each). Twenty-four cDNA libraries were constructed for Illumina sequencing, which generated a total of 86.4 Gb raw reads, with a GC% of approximately 40% per library (Appendix A). After eliminating primers, adapter sequences and low-quality reads, high-quality reads were de novo assembled into a single reference transcriptome containing 206,944 transcripts, which included 191,197 unigenes (Appendix A). BUSCO analysis indicated a transcriptome completeness of 58.3%.

### 2.3. Targeted Characterisation of Neuropeptide and Reproductive-Associated Genes

Twenty neuropeptide genes were identified from the *S. echinata* reference transcriptome (Appendix A). Based on their gene expression, the majority were exclusive to the VG of both sexes; however, *pedal peptide-2* and *NdWFamide* were only observed in the female gonad (Table 1). Several recognised oyster neuropeptide genes were not identified, including *GnRH*, *egg-laying hormone*, *elevenin*, and *NPY-1*. Only one neuropeptide gene showed significant differential expression; the *LASGLVamide* was significantly upregulated in the female pre-spawn gonad when compared to the post-spawn gonad. A total of 26 reproductive-associated genes were identified from the *S. echinata* reference transcriptome, including those showing sex-specificity (Table 2). Four genes (i.e., *SOX9*, *Dax1*, *Nanos-like*, and *Piwi-like*) were significantly upregulated in the pre-spawn female gonad, while an insulin-like peptide receptor exhibited significantly higher expression in the male VG at post-spawn stage.

### 2.4. Tissue Differentially Expressed Genes (DEGs) and Functional Annotation

#### 2.4.1. Visceral Ganglia (VG)

DEG analysis of VG transcriptomes at pre- and post-spawn stages was performed separately for female and male *S. echinata*. A total of 328 and 694 significant DEGs were identified in females and males, respectively (Appendix A). Of interest were those DEGs with significant upregulation pre- and post-spawn stages and common in both sexes. At the pre-spawn stage, both male and female VG shared 26 upregulated DEGs, of which the majority (42.3%) were uncharacterised, followed by genes associated with cell regulation and immunity (Figure 2A). Of those 26 genes, no recognised neuropeptide genes were present; however, one gene did encode a secreted uncharacterised protein (contig 1052) that was exclusive to VG. Similarly, male and female VG at the post-spawn stage consisted of 26 common upregulated DEGs (all different from the aforementioned DEGs), with the majority classified as regulatory (34.6%), followed by stress and uncharacterised genes (Figure 2B). Out of those 26 genes, three genes encoded predicted secreted proteins, including two uncharacterised proteins (contig 64031 and 19584) and a CD27 antigen-like protein (contig 197779).

#### 2.4.2. Gonads

DEG analysis of the gonad transcriptomes at pre- and post-spawn stages was performed separately for female and male *S. echinata*. In the female gonad (i.e., ovary), 661 significant DEGs were identified, including 539 upregulated in pre-spawn and 122 upregulated in post-spawn stage ovaries (Figure 3A). Those genes most highly upregulated at pre-spawn included collagen alpha-1 chain-like (contig 23914) and CD63 antigen (contig 2220), while those most highly upregulated at post-spawn included hypothetical protein (contig 145154), unknown protein (contig 155137), proteasome endopeptidase complex (contig 99954), MAG: RNA dependent RNA polymerase (contig 193181), probable ATP-dependent RNA helicase DDX46 (contig 41069), uncharacterised protein (contig 10405), and unknown protein (contig 91277). Functional annotation of all upregulated genes at both stages demonstrated an enrichment of GO terms associated with metabolic process, regulation of cellular process, transmembrane transport, biogenesis, and establishment of localization (Figure 3B,C). Notably, the pre-spawn ovary has a relatively significant abundance of expressed genes associated with signalling and signal transduction (Table 3), including receptors such as thyrotropin-releasing hormone receptor-like, metabotropic glutamate receptors, 5-hydroxytryptamine (5-HT; serotonin) receptor, tachykinin-like peptides receptor, adhesion G protein-coupled receptor L3, QRFP-like peptide receptor, and ALK tyrosine kinase receptor.

In male gonad (i.e., testis), 282 significant DEGs were identified, including 126 upregulated in pre-spawn and 156 upregulated in post-spawn testes (Figure 4A). Those genes most highly upregulated at pre-spawn included transcription elongation factor A protein 1-like (contig 18692), hypothetical predicted protein (contig 122431), and probable polyketide synthase 1 (10374), while those most highly upregulated at post-spawn included uncharacterised protein (contigs 92900, 34005, 41981) and CD63 antigen (contig 2220). Functional annotation of all upregulated genes demonstrated an enrichment of GO terms associated with metabolic process, cellular response to stimulus, regulation of cellular process, and establishment of localization (Figure 4B,C). Notably, pre- and post-spawn testes had several genes associated with signalling and signal transduction (Table 3), including receptors such as mesotocin receptor-like, allatostatin-A receptor-like, orexin receptor type, and a myosuppressin receptor-related 1-like.

## 3. Discussion

*Saccostrea echinata* are protandrous hermaphrodite oysters that undergo a complex reproductive cycle that culminates in the maturation of gametes, followed by broadcast spawning. Thus, from pre-spawn to post-spawn, there exists a massive transition in the animal’s physiology, regulated by intricate neuroendocrine pathways that have been inadequately addressed in bivalves. This study utilised targeted and comparative transcriptomics to elucidate the components that regulate the molecular transition from pre- to post-spawning events. We focused on the oyster’s organs associated with neuroendocrine controls and reproduction, i.e., the visceral ganglia and gonads, respectively. Our findings have provided an insight into the dynamics of gene expression changes before and after the spawning event, which could help optimise broodstock conditioning, enhance spawning induction success, and support successful aquaculture production of this species.

Histological examination of *S. echinata* gonads before and after a spawning event were referenced against gonad histological guidelines previously derived from *S. glomerata* [25] and *S. echinata* [16]. Our histological observations provided strong evidence for extensive cellular rearrangement of the gonads following a spawning event and, therefore, formed a foundation for tissue collection and comparative gene expression analysis.

RNA-seq platforms are now widely established and utilised to identify genes and investigate their expression, as well as associated regulatory pathways. In oysters, RNA-seq analysis has been applied to identify the genes related to reproduction, including neuropeptides that could modify broodstock gonad conditioning [21,22,26,27]. In the absence of an *S. echinata* genome or reference transcriptome for reproductive-linked investigation, we created a de novo assembly from 24 RNA-seq datasets obtained from visceral ganglia and gonads (both males and females). The assembly achieved an N50 length of 781 bp and a BUSCO completeness of 58.3%, which is within the range reported for other molluscan tissue-specific transcriptomics [27,28]. In total, our transcriptome assembly retained a total of 206,944 transcripts and 191,197 unigenes, which was comparable with the de novo assembled transcriptomes prepared for the Pearl oyster (*Pinctada margaritifera*) utilising gonad tissues, resulting in a reference transcriptome containing 70,147 unigenes [29]. Prior to the current study, only one RNA-seq dataset existed for *S. echinata,* which was derived from RNA isolated and combined from embryos/larvae and whole adult (NCBI GenBank accession number PRJNA379157) [11]. The outcome was a de novo assembled transcriptome consisting of 120,621 genes, which was used to assess orthologous sequences in other oyster species. In the current study, we take this further by performing a comparative analysis of RNA-seq data between pre-spawn and post-spawn stages to elucidate the temporal dynamics of gene expression and identified neuropeptide genes and other genes associated with this transition.

Neuropeptides act as endogenous neuroendocrine signalling factors (also includes neurotransmitters) that are synthesised and released by modified neurons, called neurosecretory cells [30]. Neuropeptides play vital roles in the neuroendocrine control of many physiological functions such as feeding, growth, and reproduction, including gametogenesis and sexual maturation [20]. We identified twenty neuropeptide precursors in *S. echinata* transcriptomes via a targeted investigation, which were mostly expressed in the VG. Previously, a total 26 neuropeptide precursors has been identified from the VG of *S. glomerata* [22], of which the *APGWamide*, *buccalin*, *crustacean cardioactive peptide*, *and LFRFamide* were implicated in gonadal maturation and spawning in both male and female *S. glomerata* [22]. We identified all of those in *S. echinata*; however, none were significantly differentially expressed between pre- and post-spawn. Of interest, the *LASGLVamide* was significantly upregulated in the pre-spawn ovary when compared to the post-spawn ovary. The *LASGLVamide* family of neuropeptides was first identified in silico in the marine gastropod *Lottia gigantea* [31]. Since then, the homologs have been predicted in *Crassostrea gigas* [21] and *Saccostrea glomerata* [22], and their mature peptides have been confirmed in *Mytilus yessoensis* [32]. Despite genomic and transcriptomic evidence of their presence, the biological function of *LASGLVamide* neuropeptides remains unknown [33], which could be further investigated. Moreover, the high expression of *LASGLVamide* in the pre-spawn ovary could be hypothesised to have a reproductive role in *S. echinata*. Further functional validation would validate the role of these genes in *S. echinata* reproduction.

Identification of genes that have been reported as reproduction-associated in other oyster species were also targeted in our study, providing comparative understanding of the potential common molecular mechanisms that control gametogenesis and spawning. Of those previously reported [22], 26 were found in the *S. echinata* reference transcriptome, which have well-established roles in different categories of reproductive processes such as germline development, gonadal development, maturation, and spawning. This highlights their potential key function in *S. echinata* reproduction. Notably, *Dax1*, *Piwi 1-like*, and *Sox1* were significantly upregulated in the pre-spawn female gonad. *Dax1*, encoded by the *NROBA* gene, is a member of the nuclear hormone receptor family [34]. The function of *Dax1* in higher vertebrate females is related to sex determination [35,36] and testicular improvement in lower vertebrates [37,38]. However, its role in molluscan gametogenesis is still poorly understood [39]. *Piwi* encodes a family of proteins that are highly evolutionarily conserved and play a crucial role in the regulation of germ cell development, gametogenesis, and self-renewal of stem cells [40]. In molluscs, the role of piwi in germline development has been well-documented by recent studies [41,42,43]. For example, in *C. gigas*, a Cg-piwi-like mRNA was expressed in both male and female diploids, with higher expression in males compared to females, which also aligns with our study. *Sox11* belongs to the *Sox* family (Sry-related high mobility group), which is a widely studied transcription factor playing a vital role in sex differentiation, gametogenesis, and vitellogenesis of fish [44]. Still, specific functions in molluscs remain largely unknown. Our findings, where *Sox11* was relatively highly expressed in pre-spawn female gonad, suggest that it could be a good target gene for further investigation into reproductive readiness in oyster reproduction.

Spermatogenesis refers to the development and proliferation of sperm cells orchestrated by dynamic gene expression that supports cell division, meiosis, and the maturation of spermatozoa [45]. In this study, several male-specific genes, such as testis-specific serine/threonine-protein kinases (*Tssk1*, *Tssk4*, *Tssk5*), spermatogenesis-associated proteins (*Spata7*, *Spata48*), and spermatogenesis and centriole-associated 1, were prominently expressed in male gonads, emphasising their role in the late stages of spermatogenesis. Previous research has shown that the *Tssk* family are exclusively expressed in the male gonad, indicating their unique role in spermatozoa function or spermatogenesis [28]. To date, five members of the *TSSK* family have been identified, each with distinct spatial and temporal expression in mammals. Targeted deletion of *Tssk1* and *Tssk2* results in disruption of spermiogenesis and male infertility [46,47]. Consistent with our findings, *Tssk1/2, Tssk3*, *Tssk4*, and *Tssk5* are dominantly expressed in the testis of various molluscs, such as the pen shell (*Atrina pectinata)*, abalone (*Haliotis discus hannai*), and Bay scallop (*Argopecten irradians*), indicating their involvement in spermatogenesis [48,49,50]. In the genome of *S. glomerata*, five candidate genes belonging to the *TSSK* family have been identified, indicating a close evolutionary relationship among these species [51]. Additionally, Boutet et al. [52] reported that spermatogenesis and centriole-associated 1 protein, or speriolin, is a testis-related protein that is expressed abundantly in the male gonad, but lower in other tissues of scallop.

The ovary is the main reproductive organ in female bivalves and is responsible for producing and maturing oocytes through a process known as oogenesis [53]. We identified several unigenes involved in oocyte maturation from the *S. echinata* transcriptome, including vitellogenin, oestrogen receptor, and nanos-like. Vitellogenesis, which is essential for female reproduction, entails the accumulation of the key yolk protein vitellin, facilitating the growth and maturation of oocytes [54,55] and can serve as a biological marker for ovarian development in both vertebrates and invertebrates [56,57]. The higher expression of vitellogenin in the ovary of *S. echinata* is consistent with results of other oyster species such as the Fujian oyster (*Crassostrea angulate)* [58], Pacific oyster (*C. gigas*) [59], and Chinese clam (*Cyclina sinensis*) [60]. Oestrogen regulates vitellogenin gene transcription in vertebrates by binding to oestrogen receptors (ER) in target organs [61]. In oyster, there is a disputes regarding the presence of E2 receptor due to the absence of aromatase [62], but later research has demonstrated a significant increase in caVg mRNA expression after E2 administration which hypothesised that E2 is primary promoter Vg mRNA transcription in *C. angulata* [58]. In our study, the presence of ERs in female gonads suggests that it could be involved in vitellogenin regulation of *S. echinata*. Gene silencing of the ER in Manila clams showed that ovarian development in females is significantly affected by decreasing ER expression, while increasing gonadal development in males [63], further supporting a role for ER in reproductive development in oysters.

Nanos proteins are well-known for their conserved role in germline development and pluripotency across various species [64,65,66]. In molluscs, *Nanos* orthologs have been observed in embryos of *Ilyanassa obsoleta* [67], *Haliotis asinina* [68], and *Pinctada fucata* [69]. In our transcriptomes, a significant up-regulation of *Nanos* gene expression was observed in the gonad of pre-spawn *S. echinata*. This is consistent with Xu et al. [70], who reported up regulation of *Nanos-like* transcripts in *C. gigas* female gonads during the early embryonic stages, indicating their role in gonad development.

Besides neuropeptides, the VG serves as a key centre for the regulation of gonad activities [22,24]. By comparing VG gene expression at pre- and post-spawn stages, we could assess common and unique genes and their potential functions in relation to the spawning event. Overall, 26 genes with significant abundance during the pre- and post-spawn stages were classified as uncharacterised, regulatory stress, and immune-related genes. Notable percentage of uncharacterised genes were reported in both pre- and post-spawn visceral ganglia for which gene characterisation and functions remain to be elucidated; however, they are flagged as having potential functions related to spawning activity in *S. echinata*. Oyster gametogenesis is believed to rely heavily on glycogen, which serves as the primary energy storage molecule to support gamete development [71,72]. The conversion of glycogen into lipids during gametogenesis generates energy for developing eggs and sperm, and upon spawning, glycogen reserves reach a minimum level [71,73]; as a result, post-spawn oysters may show increased vulnerability to thermal stress [74]. As an animal experiences increased stress, it expends more energy trying to manage that stress [75]. This results in a decreased adenylate energy charge (AEC) value, which indicates the level of metabolically available energy within an adenine nucleotide pool and serves as a vital indicator of an organism’s physiological vigour and overall health status. Li et al. [76] observed a lower level of AEC in the post-spawn *C. gigas* following the depletion of glycogen, which reduces their metabolic activity due to lower energy reserves. These findings are consistent with our study, as post-spawning male and female *S. echinata* showed no gene expression regarding metabolic activity. However, alterations of hemocyte density and phagocytosis have been observed in post-spawn oyster and do not resume as pre-spawn oyster until the next spawning event, as reported by [76]. As antimicrobial agents are synthesised and stored by hemocytes, a decrease in hemocyte density could lead to a coincidental decrease in antimicrobial activity in the haemolymph, which leads to reduced immune activity [77,78,79]. In our study, we observed a lower immune related gene expression in partially spent *S. echinata* compared to pre-spawn, which could be the reason for the reduced hemocyte density due to stress.

Gene ontology analysis of the DEGs further characterised genes that were upregulated in the pre-spawn gonads (male and female), demonstrating an abundance of genes associated with signalling and signal transduction, which is likely a requirement to facilitate the transition to and from the spawning event [28]. Several receptors, such as the tachykinin-like peptide receptor, thyrotropin-releasing hormone receptor, and a 5-HT receptor were identified. These are commonly associated with the neuroendocrine signalling of reproductive processes, including reproductive behaviour, gonad maturation, and reproductive preparedness in oysters [80,81,82]. In addition, mesotocin receptor-like and allatostatin-A receptors are known to bind to mesotocin and buccalin, respectively [83,84]. Mesotocin receptor-like belongs to the oxytocin or vasopressin receptors superfamily and plays a role in reproductive function in non-mammalian vertebrates [85]. In molluscs, the first molecular characterisation and peptide purification of oxytocin/vasopressin-type (VP/OT) neuropeptides was carried out from the neural ganglion of a pond snail (*Lymnaea stagnalis*). Later, the expression of VP/OT neuropeptides proteins isolated from the penis nerve and vas deferens of *L. stagnalis* was reported, which indicates their role in triggering the vas deferens and sperm ejaculation for *L. stagnalis* [86,87]. In this study, the presence of mesotocin in the pre-spawn testis of *S. echinata* indicated its potential role in triggering the male reproductive system. Although relevant research on the function of mesotocin or its exact homologs in oysters is lacking, the evolutionary conservation of oxytocin/vasopressin neuropeptides suggests that a potentially similar signalling system exists in oysters.

Allatostatin A (AST-A) is a member of the allatostatin (AST) neuropeptide family, which was first isolated from arthropods [88,89], and it has diverse functions as insect allatoregulatory peptides [90]. The AST-A peptides activate receptors of the rhodopsin-beta GPCR cluster to trigger the signalling mechanism as a regulatory neuropeptide [91,92,93]. Interestingly, molluscan buccalins are homologous to insect AST-A, which were first recognised and functionally characterised in sea hare (*Aplysia californica*), where they regulate muscle contraction and feeding [94]. However, recent studies have reported that buccalins have a function in reproductive activities and spawning in *S. glomerata* [22], suggesting that the presence of the AST-A receptor in our study may have a similar function in *S. echinata* spawning.

Spawning in oysters is considered a stressful event due to the significant energy expenditure related to gamete release [79]. Gene ontology analysis of the gonad DEGs showed a considerable enrichment of GO terms within “response to stress” in post-spawn females and males, which likely represents a response to stress after spawning. Notably, superoxidase dismutase (SOD) and DnaJ Homologue Subfamily A Member 1-like (DNAJA1) proteins were identified in post-spawn females’ and males’ visceral ganglia. Superoxide dismutase is an enzyme group that protects cells from oxidative stress by converting harmful superoxide radicals (O_2_^−^) into hydrogen peroxide. Research shows that the SOD gene expression varies during stress responses in bivalves such as oyster [95], mussels [96], and clams [97]. Liu et al. [98] identified SODs from the genome of *Crassostrea* and *Saccostrea* oysters and characterised them into five groups including, Mn-SODs, Cu-only-SODs, Cu/Zn ion ligand Cu/Zn-SOD with enzyme activity, Zn-only-SODs, and no ligand metal ions Cu/Zn-SODs. According to Liu, Bao, Lin, and Xue [98], most of the extracellular Cu/Zn-SOD proteins showed no enzyme activity in oysters due to the lack of conserved ligand amino acids despite of their higher levels of gene expression, which suggests their functions needs further investigation. To date, only one cytosolic Cu/Zn-SOD (*cg_XM_034479061.1*) has showed conserved enzyme activity with changing expression pattens in response to various stressors [98]. In our research, we also found a Cu/Zn-SOD in post-spawning male and female *S. echinata*, which indicates that its potential function regarding the post-spawning stress need to be elucidated. DnaJ Homologue Subfamily A Member 1-like (DNAJA1) is a member of the heat shock protein (Hsp40) family, which plays a vital role in the cellular stress response by protein refolding, trafficking, and preventing apoptosis. The specific function of DNAJA1 in molluscs is still unknown, but the upregulation in post-spawn male and females may suggest its role in the post-spawning stress recovery of *S. echinata.*

## 4. Materials and Methods

### 4.1. Experimental Animals and Sample Collection

Adult *S. echinata* at an advanced reproductive stage were obtained from the Bowen fresh oyster farm (Bowen, Northern QLD, Australia), within the period of October to March 2024. The animals were then kept in a closed recirculating saltwater system (34 ± 1 ppt and 24 ± 1 °C) at the University of the Sunshine Coast (Sippy Downs, QLD, Australia). The oysters were fed with commercial microalgae concentrate shellfish diet 1800^®^ and LPB (Reed Mariculture, Campbell, CA, USA) twice per day if they were to be kept for >1 week. Under the animal ethics committee regulations for Queensland (Australia), bivalve research does not require animal ethics approval. Nonetheless, the animals were housed and handled with care to minimise unnecessary stress. Male (149.25 ± 44.01 g) and female (180.5 ± 20.56 g) *S. echinata* were identified based on microscopic examination of the gonad, then three individuals, at an advanced reproductive stage (ripe gonad condition; pre-spawn) and at 2 h after spawning (partially spent gonad condition; post-spawn) were selected, totalling 12 animals. Gonad and VG were dissected from each oyster and immediately frozen in liquid nitrogen before storage at −80 °C until use. In addition, a 6 mm thick cross-section of the visceral mass at the level of the gill and labial pulp junction was collected and fixed in 4% paraformaldehyde in phosphate-buffered saline (PBS) for 24 h. Each sample was then washed with 1XPBS for 15 min and preserved in 70% ethanol until use for histology.

### 4.2. Gonad Histology

Fixed gonad tissues were processed through a graded series of ethanol to 100%, then embedded in paraffin before sectioning into 5-μm-thick sections [25]. Sections were then processed through hematoxylin and eosin staining, followed by routine hematoxylin and eosin (H&E) protocol [16]. The gonad sections were assessed microscopically to determine sex and gonadal development according to [26], which includes inactive, early active, late active, ripe, spawning, partially spent, and spent stages. Microscopic images were taken using a Leica compound microscope DM5500 and DMC6200 camera (Leica Microsystems, Wetzlar, Germany).

### 4.3. RNA Extraction, Library Construction, and RNA Sequencing

Total RNA from gonads and VG at pre- and post-spawn stage *S. echinata* was extracted using TRIzol reagent (Invitrogen, Mount Waverley, VIC, Australia), as per the manufacturer’s protocol. Extracted RNA was assessed for quality by visualisation on a 1.2% agarose gel and quantified using a Nanodrop spectrophotometer 2000c (Thermo Scientific, Waltham, MA, USA) at wavelengths 260 and 280 nm. Total RNA was stored at −80 °C until further use. Twenty micrograms of total RNA from each sample were dried in RNA storage tubes (GenTegra, Pleasanton, CA, USA) and sent to Novogene (Singapore) for library construction and sequencing (150 nucleotides paired-end reads) using a Next-Generation Illumina NovaSeq PE150 platform, according to the standardised Illumina pipeline “(https://en.novogene.com, accessed on 25 June 2024)”. RNA samples with an RNA integrity number ≥ 6 were sequenced. All RNA-seq raw data was submitted to the NCBI GeneBank under accession number PRJNA1236486.

### 4.4. Reference Transcriptome Assembly and Annotation

Before assembly, the raw Illumina reads in high-throughput sequencing were converted to FASTQ format, trimmed to remove the adapter sequences, duplicated sequences, ambiguous reads, and low-quality reads to ensure the accuracy of the subsequent analysis. Due to the lack of genome information on *S. echinata*, clean reads from eight libraries generated in this study were used for a de novo assembly using the CLC Genomics Workbench software version (21.0.3), with the default parameter settings (minimum contig length, 20, mismatch cost, 2.0, insertion cost, 3.0, length fraction, 0.5). Prior to a de novo assembly, a library quality check followed by the removal of low-quality reads was performed. After concatenating the assembled transcripts into a single file, Omics Box software v 2.1.2 (BioBam Bioinformatics S.L.) was used to run the CD-HIT (version 4.8.1) with the setting of “-c 0.95” to remove transcripts whose sequence similarity exceeded 95% [99]. The resulting sequences were called unigenes. Analysis of assembly completeness was performed using BUSCO (Version 5.4.5) to obtain the percentage of single-copy orthologs represented in the mollusca_odb10 dataset [100]. Open reading frames (ORFs) of transcript and unigene sequences were predicted by TransDecoder (version 5.5.0), with the minimum ORF length of 100 bp [101]. To obtain the biological functional annotation relevant to *S. echinata,* BLASTx was performed using Omics Box software (version 2.1.2, BioBam Bioinformatics S.L. Valencia, Spain) against the NCBI (http://www.ncbi.nlm.nih.gov/, accessed on 1 October 2024) non-redundant protein database for molluscs to obtain biological functional annotation relevant to *S. echinata*. Matches with an e-value < 10^−5^ and an annotation score of 45 were considered significant hits, and the best reciprocal matches were selected based on their scores [102]. To increase the annotation quality, InterProScan (version 5.72-103.0), a protein-family-based annotation tool, was employed to search for all Pfam protein family profiles [103]. Finally, in the gene function analysis, we used those proteins that were blasted from the databases. We did not consider the unannotated, long non-coding RNA sequence for gene ontology since their functions were not available.

### 4.5. Targeted Identification of Neuropeptide and Reproductive-Associated Genes

Molluscan neuropeptide and reproduction-associated gene sequences, previously documented in the literature [22], were queried in tBLASTn searches to identify homologous sequences in the *S. echinata* De Novo transcriptome using the CLC Genomics Workbench software version (21.0.3) with e-value cut-off < 10^−5^. The gene expression value in transcripts per million (TPM) of the identified sequences was retrieved after mapping each RNA-Seq dataset with the generated *S. echinata* de novo assembled transcriptome. Genes with TPM values below 5 were excluded from this study.

### 4.6. Differential Gene Expression Analysis and Annotation

The cleaned reads obtained from each RNA-seq were mapped to the de novo assembled transcripts, and differentially expressed genes (DEGs) between pre- vs. post-spawn gonad and VG from each sex (*n* = 3 as per each tissue, stage, and sex) were conducted using the default parameters of CLC Genomics Workbench (version 21.0.3). Significantly upregulated or downregulated genes (FDR *p*-value < 0.05) were identified, and a volcano plot mapping differentially expressed genes (log 2 fold-change with cut-off >2.0 and FDR *p*-value < 0.05) was generated using VolcaNoseR software (https://huygens.science.uva.nl/VolcaNoseR/, accessed on 12 October 2025) [104]. To further investigate GO (Gene Ontology) terms enrichment, DEGs were compared against the non-redundant protein databases (e-value cut-off: 10^−3^) through BLASTp at the National Centre for Biotechnology Information (NCBI) using OmicsBox software version (2.1.2, BioBam Bioinformatics S.L., Valencia, Spain).

## 5. Conclusions

This research highlights the distinct and common molecular signatures associated with pre-spawning and post-spawning in oysters, utilising cultured *S. echinata*. Genes associated with gonad maturation, spawning, and neuroendocrine signalling were highlighted during the pre-spawn stage, reflecting their potential role in reproductive readiness before the release of gametes during a spawning event. Conversely, genes associated with a stress response, immune-suppression, and metabolic activity were significantly abundant at the post-spawn stage, suggesting that a spawning event triggered stress responses. Genes that were highly abundant at pre- and post-spawn stages provides valuable insights regarding the reproductive stage or spawning readiness of this species, which supports future research in broodstock management and selective breeding in *S. echinata* aquaculture systems.

## Figures and Tables

**Figure 1 ijms-26-10032-f001:**
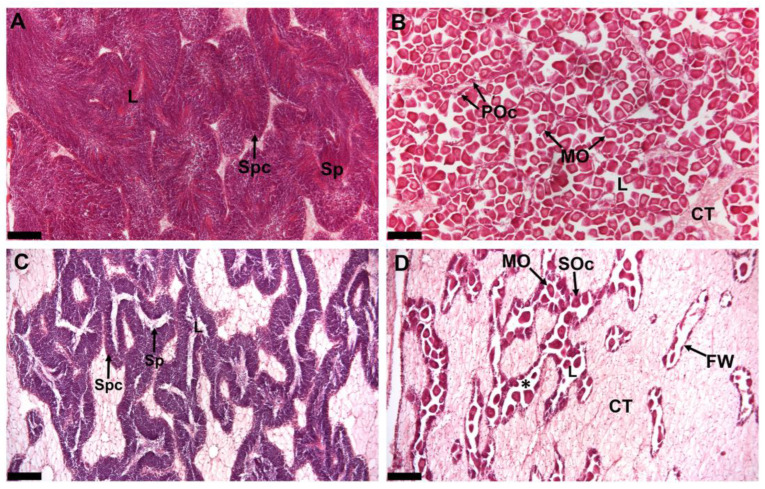
Histological sections of *S. echinata* male and female gonads. (**A**) Mature testis with gonadal follicles occupying a large area of the gonad, an absence of interstitial connective tissue, and a lumen filled with spermatozoa. Spermatocytes are located near the follicular wall. (**B**) Mature ovary packed with fully matured ova and a thin layer of interstitial connective tissue. In the germinal epithelium, small primary oocytes are present. (**C**) Partially spawned testis with vacant spaces. (**D**) Partially spawned ovary with loosely packed ova and vacant spaces due to the absence of mature oocytes in the lumen of the follicle. Advanced stage secondary oocytes that are attached to the follicular wall can be observed. Interstitial connective tissue becomes prominent. Spc—Spermatocytes, Sp—Spermatozoa, L—Lumen, MO—Mature ova, POc—Primary oocytes, CT—Connective tissue, SOc—Secondary oocytes, *—Vacant space, and FW—Follicular wall. Scale bars: 100 µm. For high magnification images, see Appendix A.

**Figure 2 ijms-26-10032-f002:**
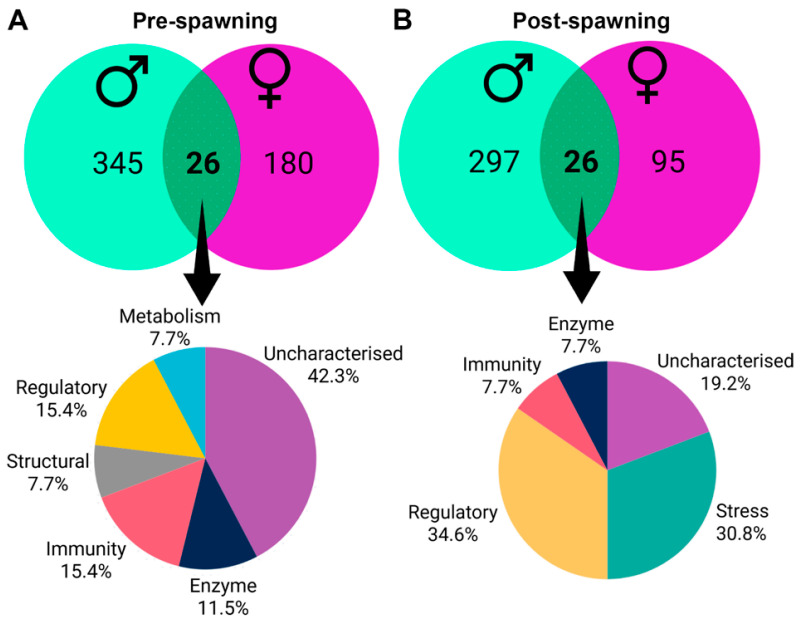
Differential gene expression between female and male *S. echinata* visceral ganglia at pre-and post-spawn. (**A**) Venn diagram showing the common and unique upregulated genes at pre-spawn between female and male, and pie chart demonstrating functional annotation of common genes. (**B**) Venn diagram showing the common and unique upregulated genes at post-spawn between female and male, and a pie chart demonstrating functional annotation of common genes.

**Figure 3 ijms-26-10032-f003:**
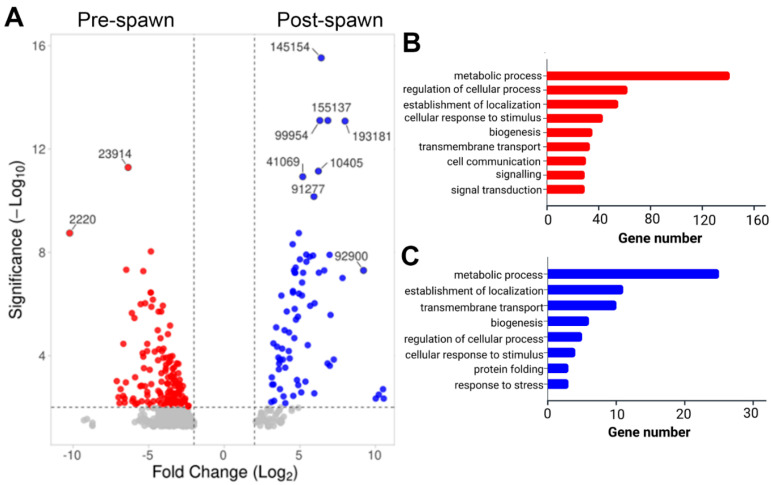
Differential gene expression analysis in *S. echinata* pre- and post-spawn female gonad (ovary). (**A**) Volcano plot showing overall significantly differential gene expression. For a summary of all genes, see Appendix A. Gene IDs are included for those at highest fold-change and/or significance. (**B**) Bar chart representing functional annotation categories for genes upregulated in the pre-spawn ovary. (**C**) Bar chart representing functional annotation categories for genes upregulated in the post-spawn ovary.

**Figure 4 ijms-26-10032-f004:**
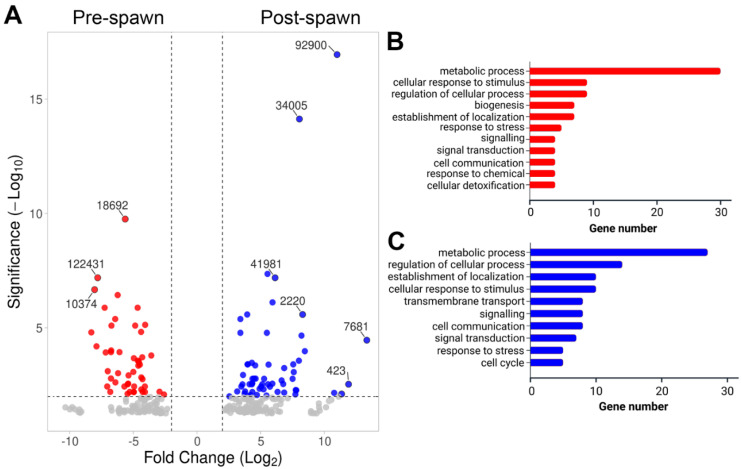
Differential gene expression analysis in male *S. echinata* pre- and post-spawn gonad (i.e., testis). (**A**) Volcano plot showing overall significantly differential gene expression. For a summary of all genes, see Appendix A. Gene IDs are included for those at highest fold-change and/or significance. (**B**) Bar chart representing functional annotation categories for genes upregulated in pre-spawn testis. (**C**) Bar chart representing functional annotation categories for genes upregulated in post-spawn testis.

**Table 1 ijms-26-10032-t001:** Summary of neuropeptide genes identified in the *S. echinata* reference transcriptome and their presence (●) in visceral ganglia (VG) and gonad. Asterisk indicates a significant upregulation at pre-spawn stage. See Appendix A for corresponding protein sequences.

Contig ID	Gene Name	Female VG	Male VG	Female Gonad	Male Gonad
146852	*Allatotropin*	●	●		
140434	*APGWa*	●	●		
46225	*Buccalin*	●	●		
71066	*CCAP-1*	●	●		
91581	*CCAP-2*	●	●		
120340	*CCK*	●	●		
136993	*Conopressin*	●	●		
113456	*GGNG*	●	●		
98593	*LASGLVamide*	●	●	● *	
86141	*LFRFa*	●	●		
74205	*LFRYa*	●			
51782	*LRNFVamide*	●	●		
86374	*Luqin*	●	●		
139379	*Myomodulin*		●		
39729	*NdWFamide*			●	
137757	*Pedal peptide-1*	●	●		
98593	*Pedal peptide-2*	●	●	●	
80267	*PKYMDT*	●	●		
125782	*Pyrokinin*	●	●		
86374	*Wwamide*	●	●		

***** Significant differences in the genes were based on (FDR *p*-value < 0.05).

**Table 2 ijms-26-10032-t002:** Summary of reproductive-associated genes identified in the *S. echinata* reference transcriptome, including their presence (●) in visceral ganglia (VG) and gonads. See Appendix A for corresponding protein sequences.

Contig ID	Gene Name	Female VG	Male VG	Female Gonad	Male Gonad
Gonad development					
448	*Adenosine deaminase-like*			●	●
5174	*Calcineurin-beta*	●	●	●	●
2311	*Catenin-beta*	●	●	●	●
41577	*Cytidine deaminase-like (CDA)*	●	●		
2102	*Dax1*	●	●	● *	●
34255	*Forkhead box protein L2-like (Foxl2)*			●	
13580	*Insulin-like growth factor-binding protein 7 (IGFBF7)*	●	●	●	●
19979	*Insulin-like peptide receptor (IR)*		● ^#^		
1086	*Piwi 1-like*			● *	●
7116	*Prohibitin 1-like*	●	●	●	●
1911	*Prohibitin 2-like*	●	●	●	●
2769	*Transcription factor SOX-11-like (Sox11)*			● *	
24745	*Transcription factor SOX-9-like (Sox9)*	●	●	●	
1218	*Transforming growth factor-beta*			●	●
150	*Vasa*				●
Spermatogenesis					
80733	*Spermatogenesis and centriole associated 1 (SCA)*				●
89650	*Spermatogenesis-associated protein 48 (SPATA48)*				●
56798	*Spermatogenesis-associated protein 7-like (SPATA7)*				●
103099	*Testis-specific serine/threonine-protein kinase 1-like (Tssk1)*				●
21110	*Testis-specific serine/threonine-protein kinase 4-like (Tssk4)*				●
88892	*Testis-specific serine/threonine-protein kinase 5-like (Tssk5)*				●
Ovarian development					
56406	*Oestrogen receptor (ER)*	●		●	
22711	*Nanos 1- like*			● *	
17	*Vitellogenin (Vg)*			●	

***** Significantly upregulated at pre-spawn stage based on (FDR *p*-value < 0.05). **#** Significantly upregulated at post-spawn stage based on (FDR *p*-value < 0.05).

**Table 3 ijms-26-10032-t003:** Summary of DEGs upregulated in female and male *S. echinata* gonad (i.e., ovary and testis, respectively) that are associated with signalling and signal transduction.

Sex and Stage	Contig ID	Description	Log2 FC (TPM)
Female Pre-spawn	47778	Thyrotropin-releasing hormone receptor-like	5.75
	32718	Guanine nucleotide-binding protein G(s) subunit alpha-like	4.68
	5974	Metabotropic glutamate receptor 3-like	4.28
	28837	Tyrosine-protein kinase transmembrane receptor Ror2-like	3.90
	15516	ADAM 17-like protease	3.74
	26931	Metabotropic glutamate receptor 8	3.72
	39055	E3 ubiquitin-protein ligase UBR5-like isoform X1	3.55
	59795	Uncharacterised protein LOC133181846	3.47
	25097	Probable nuclear hormone receptor HR38	3.43
	18453	Metabotropic glutamate receptor 3-like	3.30
	3895	Metabotropic glutamate receptor 3	3.29
	3692	Protein timeless-like	3.10
	13401	ALK tyrosine kinase receptor-like isoform X2	3.10
	2417	Guanine nucleotide-binding protein G(s) subunit alpha-like	3.09
	39896	Dual specificity protein phosphatase 1-like	3.05
	63391	5-hydroxytryptamine receptor	3.05
	22655	Tachykinin-like peptides receptor 99D	3.03
	11835	Proline-rich protein 5-like isoform X2	3.00
	5768	Hypoxia up-regulated protein 1-like isoform X1	2.94
	28224	ALK tyrosine kinase receptor-like	2.83
	15622	Activin receptor type-1-like isoform X2	2.70
	9750	Cell division control protein 6 homologue	2.63
	9970	Serine/threonine-protein kinase SIK1-like	2.53
	11308	Neurogenic locus notch homologue protein 1-like isoform X2	2.50
	20191	Phosphatidylinositol 4,5-bisphosphate 3-kinase catalytic subunit alpha isoform-like	2.38
	8892	Adhesion G protein-coupled receptor L3-like isoform X1	2.16
	20967	Glypican-6-like	2.14
	23152	QRFP-like peptide receptor	2.13
	1932	Repulsive guidance molecule A-like	2.11
Male Pre-spawn	44229	Mesotocin receptor-like	6.74
	32219	Protein immune deficiency	5.02
	67392	Allatostatin-A receptor-like	4.86
	152920	Protein IMPACT homologue	4.69
	64014	Uncharacterised protein LOC133197027	4.14
	83878	Allatostatin-A receptor-like	3.33
	33855	Death domain-containing protein CRADD-like	3.20
	23863	Uncharacterised protein LOC133197027	2.45
Male Post-spawn	64930	Tumour necrosis factor ligand superfamily member 11 isoform X1	2.54
	8051	Orexin receptor type 2-like	3.23
	16500	Ras-related protein Rab-39B-like	3.98
	27769	G-protein coupled receptor dmsr-1-like	4.48

## Data Availability

Dataset available on request from the authors.

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
