# Peer review of "Comparative Transcriptomics Provides Insight into the Neuroendocrine Regulation of Spawning in the Black-Lip Rock Oyster (Saccostrea echinata)"

_ijms, 2025, doi:10.3390/ijms262010032_

Round 1
Reviewer 1 Report
Comments and Suggestions for Authors
Manuscript: ijms-3877825
Insights into the neuroendocrine regulation of spawning in the black-lip rock oyster (Saccostrea echinata), by Zafar et al.
The manuscript explores neuroendocrine regulation of spawning in Saccostrea echinata through histological examination and comparative transcriptomics of gonads and visceral ganglia at pre- and post-spawn stages.
General comments:
The study addresses an important aquaculture species and provides valuable baseline molecular information. The RNA-seq data are of good technical quality and the integration with histology adds relevance. However, the biological interpretation is limited and probably the authors need to made clear about the limitations caused by the presence of many uncharacterized genes of uncertain functional relevance.
While the work offers useful descriptive insights clearer focus on biologically meaningful pathways would be needed to significantly advance understanding of reproductive regulation in this species.
Related to histology, although the sections and images allow for general differences between the groups studied, it would be good to include insets at higher magnification to appreciate the structures or to add a figure with the photos at higher magnification as supplementary material.
A major concern is the inclusion of large numbers of uncharacterized or hypothetical proteins in the results and discussion. While it is expected that de novo transcriptome assemblies will generate many such entries, the manuscript repeatedly highlights these genes as part of the main findings (e.g., “hypothetical predicted protein – Line 204” upregulated in post-spawn tissues). These results add little to no value for the reader, as they do not provide any biological insight, not even from a descriptive perspective. The authors should carefully reconsider the relevance of reporting such genes as key findings, or at minimum, restrict their presentation to supplementary material until their potential role can be functionally annotated. Otherwise, the emphasis on poorly characterized sequences risks diluting the impact of the manuscript and may distract from the more meaningful biological signals.
The discussion links the genes found with known reproductive processes, but without direct functional evidence. Functions are assumed based on orthologs in other mollusks, which is valid as a hypothesis, but insufficient for such conclusive statements. Examples: Lines 364-366, 384-386.
Include a statistical analysis section in materials and methods detailing what has been done in the work
Although my native language is not English, I identify issues of grammar and spelling that should be carefully reviewed throughout the manuscript.
Specific comments:
Line 69. Replace “neuropeptides; neuropeptides are” with “neuropeptides. These compounds”.
Line 106 (and others like 128, 135, 146): use italics for the species name.
Line 115: The asterisks (*) cannot be seen in the image.
Line 126: Table 1 could be a supplementary table; because the data is repeated with the 1st paragraph of 2.2. In table 1, delete the bullet points of “Single-copy BUSCO” and “Duplicated BUSCO”
Line 131: What interpretation do the authors make of the failure to identify known neuropeptides in molluscs such as GnRH, egg-laying hormone, elevenin, NPY-1 or APGWamide?
Line 142: Include the reference value taken to indicate whether or not there are significant differences, as well as the statistical test used.
Line 147: As recommended for Table 2, include the reference value taken to indicate whether or not there are significant differences, as well as the statistical test used.
Lines 187-190: What does it mean for the authors that these genes have been identified?
Line 204: What is an “hypothetical predicted protein”?
Line 229-230: On what evidence do you claim that it provides such a function?
Line 233-235: What is new and what is the strong result that you have found in this organism?
Line 264: Apparently something is missing “Of interest, the was significantly upregulated…”
Line 325: What does this mean? “in caVg mRNA”
Line 392: “functionally”
Line 404: Replace “less toxic molecules” with “hydrogen peroxide”
Line 404: Delete “the expression of”
Lines 414-416: The authors just found it there? How do you explain that the only way to eliminate the superoxide anion isn't expressed in the other condition?
I hope that the comments of the manuscript will be constructive for the authors and I encourage them to make an appropriate review of the suggestions to improve the quality of the manuscript.
Author Response
General comments:
Comment: While the work offers useful descriptive insights clearer focus on biologically meaningful pathways would be needed to significantly advance understanding of reproductive regulation in this species.
Response: We appreciate the reviewer’s perspective; however, our study was designed as a foundational analysis to establish candidate genes and neuropeptides involved in reproductive regulation of S. echinata. This species has relatively little foundational understanding compared to the Pacific oyster, Crassostrea gigas. Thus, we believe this descriptive framework provides essential groundwork that will enable more targeted pathway-focused investigations in future studies.
Comment: Related to histology, although the sections and images allow for general differences between the groups studied, it would be good to include insets at higher magnification to appreciate the structures or to add a figure with the photos at higher magnification as supplementary material.
Response: We thank the reviewer for this constructive suggestion. We have now provided additional higher magnification images as insets to Figure 1.
Comment: A major concern is the inclusion of large numbers of uncharacterized or hypothetical proteins in the results and discussion. While it is expected that de novo transcriptome assemblies will generate many such entries, the manuscript repeatedly highlights these genes as part of the main findings (e.g., “hypothetical predicted protein – Line 204” upregulated in post-spawn tissues). These results add little to no value for the reader, as they do not provide any biological insight, not even from a descriptive perspective. The authors should carefully reconsider the relevance of reporting such genes as key findings, or at minimum, restrict their presentation to supplementary material until their potential role can be functionally annotated. Otherwise, the emphasis on poorly characterized sequences risks diluting the impact of the manuscript and may distract from the more meaningful biological signals.
Response: We appreciate the reviewer’s concern regarding the inclusion of uncharacterized or hypothetical proteins in the results and discussion. We fully acknowledge that the biological function of these transcripts remains unclear at present, which is a common outcome of de novo transcriptome assemblies in non-model species such as this oyster species. However, we believe it is important to report these genes, as they may represent novel or lineage-specific candidates that could play previously unrecognised roles in reproduction. Highlighting their presence provides a baseline for future functional studies and ensures that potentially relevant signals are not overlooked.
Comment: The discussion links the genes found with known reproductive processes, but without direct functional evidence. Functions are assumed based on orthologs in other mollusks, which is valid as a hypothesis, but insufficient for such conclusive statements. Examples: Lines 364-366, 384-386.
Response: We thank the reviewer for this valuable comment. We agree that our functional interpretations are based on orthology with genes characterised in other molluscs and related taxa. To address this, we have discussed that the need for future functional studies to validate the causal role of these genes in oyster reproduction.
Comment: Include a statistical analysis section in materials and methods detailing what has been done in the work
Response: Thanks. The detailed statistical analysis has been discussed in “Materials and Methods” section under the subheading of “Differential gene expression analysis and annotation”.
Comment: Although my native language is not English, I identify issues of grammar and spelling that should be carefully reviewed throughout the manuscript.
Response: We thank the reviewer for this observation. The entire manuscript has been carefully reviewed for grammar, spelling, and clarity, and has been polished to improve readability.
Specific comments:
Comment: Line 69. Replace “neuropeptides; neuropeptides are” with “neuropeptides. These compounds”.
Response: We agree with this comment. Therefore, we have updated the manuscript with “neuropeptides. These compounds” in the “Introduction” section, Page number 02.
Comment: Line 106 (and others like 128, 135, 146): use italics for the species name.
Response: We agree with this comment. Therefore, we have updated the manuscript with “italics with the species name” in the “Introduction” and “Results” section.
Comment: Line 115: The asterisks (*) cannot be seen in the image.
Response: We agree and have updated the asterisks to be visible and bigger on the images. Figure 1 on the “Results” section.
Comment: Line 126: Table 1 could be a supplementary table; because the data is repeated with the 1st paragraph of 2.2. In table 1, delete the bullet points of “Single-copy BUSCO” and “Duplicated BUSCO”.
Response: We agree with this comment. Therefore, we have deleted the “bullet points” from Table 1 and make it as a supplementary table (Table S2).
Comment: Line 131: What interpretation do the authors make of the failure to identify known neuropeptides in molluscs such as GnRH, egg-laying hormone, elevenin, NPY-1 or APGWamide?
Response: Thanks for pointing this out. For your information, we identified APGWamide in both male and female VG (Table 2, Page number 05) of our animals. The failure to identify of GnRH, egg-laying hormone, elevenin, NPY-1 neuropeptides could reflect genuine lineage-specific differences in neuropeptide repertoires between oyster species and other molluscs. Alternatively, their absence may result from technical factors, such as sequence divergence leading to low homology-based detection, incomplete genome/transcriptome coverage, or expression below the detection threshold in the sampled tissues and developmental stages.
Comment: Line 142: Include the reference value taken to indicate whether or not there are significant differences, as well as the statistical test used.
Response: We apologise that we did not mention the reference value of significant differences. We updated the manuscript with the value which is in the “Results” section (as a Table legend) in Table 2.
Comment: Line 147: As recommended for Table 2, include the reference value taken to indicate whether or not there are significant differences, as well as the statistical test used.
Response: Thanks again for pointing this out. We updated the manuscript with the value which is in the “Results” section (as a Table legend) in Table 3.
Comment: Lines 187-190: What does it mean for the authors that these genes have been identified?
Response: The identification of these signalling- and receptor-related genes indicates that the ovary is an active signalling environment, responsive to neuropeptides (e.g., tachykinin-like peptides, QRFP-like peptides), neurotransmitters (serotonin, glutamate), and hormones (thyrotropin-releasing hormone-like) during early gametogenesis of Black-lip rock oyster.
Comment: Line 204: What is an “hypothetical predicted protein”?
Response: A hypothetical predicted protein refers to a computationally annotated sequence that is presumed to encode a protein but lacks experimental validation or known function.
Comment: Line 229-230: On what evidence do you claim that it provides such a function?
Response: This conclusion is supported by our differential gene expression analysis and functional annotation, which revealed a clear shift in the expression of neuropeptide, receptor, and signalling genes between pre-spawn and post-spawn oysters.
Comment: Line 233-235: What is new and what is the strong result that you have found in this organism?
Responses: This is the first histological observation related to pre- and post-spawn gonad of S. echinata and the clear cellular changes/rearrangement indicates the strong evidence of right gonad stages during tissue collection.
Comment: Line 264: Apparently something is missing “Of interest, the was significantly upregulated…”
Response: Thanks for pointing this out. We have added the missing word which is “LASGLVamide” in “Discussion” section.
Comment: Line 325: What does this mean? “in caVg mRNA”
Response: Thanks for your comments. The short form of “ca” is Crassostrea angulata, “Vg” = Vitellogenein
Comment: Line 392: “functionally”
Response: We updated the typo mistakes in “Discussion” section.
Comment: Line 404: Replace “less toxic molecules” with “hydrogen peroxide”
Response: We updated with “hydrogen peroxide” in the “Discussion” section.
Comment: Line 404: Delete “the expression of”
Response: We have deleted “the expression of” Line number 406, Page number 14.
Comment: Lines 414-416: The authors just found it there? How do you explain that the only way to eliminate the superoxide anion isn't expressed in the other condition?
Response: Thanks for your comment. We found that this superoxide (Cu/Zn-SOD) in post-spawn male and female. The elevated expression observed in post-spawn oysters likely reflects increased oxidative metabolism and tissue remodelling following gamete release.
Reviewer 2 Report
Comments and Suggestions for Authors
The manuscript provides novel insights into the molecular mechanisms underlying gonad maturation and spawning regulation in Saccostrea echinata, an emerging aquaculture species. By integrating histological analysis with transcriptomic profiling of gonads and visceral ganglia, the authors identify key neuropeptide and reproduction-related genes associated with pre- and post-spawning stages. This work has high potential value for aquaculture management and selective breeding strategies. The study is timely, well-structured, and presents important findings, but several sections would benefit from further clarification, methodological detail, and contextualization within broader bivalve reproductive biology.
Specicif comments:
- Introduction
The introduction clearly establishes the importance of oyster aquaculture and the need to understand reproductive regulation in S. echinata. However, the rationale for focusing on neuroendocrine regulation could be expanded by highlighting knowledge gaps compared to other oyster species (Crassostrea gigas, Saccostrea glomerata).
- Materials and Methods
The methodology for histological staging is well presented, but details on the sampling timing should be clarified to ensure reproducibility.
- Results
The identification of neuropeptides is central to the paper. However, the biological interpretation of why LASGLVamide is upregulated only in pre-spawn female gonad requires expansion. Are there known homologs with reproductive roles in molluscs?
Results on reproductive-associated genes (SOX9, Dax1, Nanos-like, Piwi-like) are compelling but should be compared more explicitly with previous findings in other oysters. A comparative summary table across species would strengthen the study.
DEG results contain a large number of uncharacterised proteins. Functional annotation of these should be emphasised, perhaps by focusing on the most promising candidates for reproductive control.
- Discussion
Consider addressing more directly how these findings may help overcome challenges in S. echinataaquaculture (low larval survivability, poor spawning induction).
The interpretation of stress-related gene expression (SOD, DNAJA1) in post-spawn stages is interesting, but the authors should more critically evaluate whether these responses are species-specific or reflect general molluscan stress physiology.
- References
Try to cite some newly published articles, such as https://doi.org/10.1016/j.agrcom.2023.100016
- Language and Grammar
The manuscript is well-written but occasionally verbose.
Ensure consistent use of scientific terms.
Author Response
Comment: Introduction – The introduction clearly establishes the importance of oyster aquaculture and the need to understand reproductive regulation in S. echinata. However, the rationale for focusing on neuroendocrine regulation could be expanded by highlighting knowledge gaps compared to other oyster species (Crassostrea gigas, Saccostrea glomerata).
Response: We thank the reviewer for this helpful suggestion. In the revised “Introduction”, we have expanded the rationale and highlighted the knowledge gaps.
Comment: Materials and Methods – The methodology for histological staging is well presented, but details on the sampling timing should be clarified to ensure reproducibility.
Response: Thanks for pointing this out. We added the sample collection time in the “Materials and methods” section under the subheading of “Experimental animals and sample collection”
Comment: Results – The identification of neuropeptides is central to the paper. However, the biological interpretation of why LASGLVamide is upregulated only in pre-spawn female gonad requires expansion. Are there known homologs with reproductive roles in molluscs?
Response: Thanks for pointing this out. The homolog of the LASGLVamide has been predicted in Crassotrea gigas [21] and Saccostrea glomerata [22] and their mature peptide has been confirmed in Mytilus yessoensis [32], which we discussed on the Discussion section. However, we also expand the biological interpretation of highly upregulated LASGLVamide in pre-spawn ovary, which is: “Moreover, the high expression of LASGLVamide in the pre-spawn ovary could be hypothesized to have a reproductive role in S. echinata” in the “Discussion” section.
Comment: Results – Results on reproductive-associated genes (SOX9, Dax1, Nanos-like, Piwi-like) are compelling but should be compared more explicitly with previous findings in other oysters. A comparative summary table across species would strengthen the study.
Response: We thank the reviewer for this suggestion. We discussed those genes and compared insights across oyster species in the “Discussion” section. Given that only a small number of genes are involved, we believe these comparisons are best integrated into the narrative rather than presented in a separate table. We feel this approach avoids redundancy and maintains the flow of the Discussion.
Comment: Results – DEG results contain a large number of uncharacterised proteins. Functional annotation of these should be emphasised, perhaps by focusing on the most promising candidates for reproductive control.
Response: Due to lack of knowledge, it is difficult to gain any further insights into which specific uncharacterised genes are of most relevant to reproduction. Only with further investigation of their functional activity or comparative investigation with other oyster species, will give the required information. Therefore, we did not want to provide emphasis on the uncharacterised genes.
Comment: Discussion – Consider addressing more directly how these findings may help overcome challenges in S. echinata aquaculture (low larval survivability, poor spawning induction).
Response: Thanks for your comment. We updated the discussion according to the comment, which is “which could help optimise broodstock conditioning, enhance spawning induction success, and support successful aquaculture production of this species” in the “Discussion” section.
Comment: Discussion – The interpretation of stress-related gene expression (SOD, DNAJA1) in post-spawn stages is interesting, but the authors should more critically evaluate whether these responses are species-specific or reflect general molluscan stress physiology.
Response: We appreciate the reviewer’s suggestion. In the “Discussion”, we have clarified that SOD and DNAJA1 expression patterns in post-spawn S. echinata, which reflects the general molluscan stress physiology.
Comment: References – Try to cite some newly published articles, such as https://doi.org/10.1016/j.agrcom.2023.100016.
Response 8: Specific lack of newly published article is a reflection on the lack of relevant research in this specific area.
Comment: Language and Grammar – The manuscript is well-written but occasionally verbose. Ensure consistent use of scientific terms.
Response : We thank the reviewer, and we have reassessed the entire text so that the manuscript is more succinct.
Reviewer 3 Report
Comments and Suggestions for Authors
General comments
This paper tries to investigate the neuroendocrine regulation of spawning in the black-lip rock oyster, Saccostrea echinata, an emerging aquaculture species. Researchers aimed to understand the molecular mechanisms driving reproductive processes by analysing gene expression in the visceral ganglia and gonads of male and female oysters at both pre-spawn and post-spawn stages. The study highlights specific genes and signalling molecules that are differentially expressed during these reproductive phases, identifying candidates involved in gonad maturation, spawning readiness, and the subsequent stress response. Finally, these findings offer valuable insights that could lead to improved aquaculture management and breeding strategies for this oyster species.
The objective of the manuscript, in general, is interesting from aquaculture management and breeding strategies perspective. However, the manuscript requires some improvements to meet publication standards. The manuscript should become acceptable for publication pending suitable minor revision considering the comments appended below.
More specific comments:
Title: The current title is effective and acceptable, but it can be improved to frame the study's contribution more impactfully. This can be done by highlighting the methods used (e.g., Transcriptomic and Histological Analysis) or by emphasizing the core finding (e.g., the changes in gene expression and the temporal nature of the research).
Abstract: The abstract effectively summarizes the paper's key findings but can be improved for clarity. The main suggestion is to start with a more direct statement of the research problem—the challenges in managing black-lip rock oyster spawning—before detailing the study's purpose. The abstract should also condense the list of specific genes into broader categories to improve flow, while still highlighting key findings like the upregulation of the LASGLVamide gene in pre-spawn females and stress-response genes in post-spawn oysters. Finally, the conclusion could be rephrased to be more impactful by directly stating that the findings provide a molecular foundation for improving aquaculture and breeding strategies. These revisions would make the abstract easier for readers to quickly grasp the study's core contributions.
Introduction:
Line 43 – 45: "To address these issues, the NSW oyster industry is actively advancing through selective breeding for both species [3-6]". The authors could briefly explain what "advancing through selective breeding" entails and what traits are being selected for (e.g., disease resistance, faster growth rates). This would give the reader a better understanding of the applied context.
Line 48 – 50: "The black-lip rock oyster (Saccostrea echinata) is a prominent aquaculture species that has recently drawn researchers' attention as a new aquaculture commodity in Australia [9]". The introduction would benefit from a more explicit statement on why this species is of particular interest. What specific characteristics make it a "prominent aquaculture species"? Mentioning traits like fast growth, good taste, or hardiness would add valuable context.
Line 75 – 77: "It is the visceral ganglia (VG), located in close association with the posterior adductor muscle, that controls the visceral mass, including the gonad [24]". The grammatical structure is not correct. "That controls" makes it unclear whether the muscle or the ganglia controls the visceral mass. Please revise
Line 84 – 87: "In this study, to better understand ... and reproduction-associated genes". This is a long, run-on sentence. It combines the overall objective with the specific methods. try to split it into two sentences for better flow and clarity. Also, while the objectives are present in this paragraph, they are embedded in prose. Try to reformat the final paragraph to clearly state the study's aims.
General comment: The flow from the discussion of aquaculture challenges to the specific focus on neuroendocrine regulation could be smoother. Try to improve the transition between paragraphs after discussing the challenges of mass mortality and selective breeding to directly link the problem to the research question.
Results:
The transcriptome assembly statistics are presented in Table 1, including N50 length and BUSCO completeness. While these numbers are provided, a breif discussion of what these metrics mean in the context of the study would be beneficial. For instance, what does a BUSCO completeness of 58.3% imply for the study's ability to find all relevant genes? Please add a sentence or two to the beginning of the results section that interprets the assembly statistics.
Figure 1 shows the histological sections, but the description is brief. In the caption for Figure 1, or in the text describing the figure, please provide a more detailed description of the key histological features for each stage.
Discussion:
The discussion mentions the differential expression of several receptors (e.g., thyrotropin-releasing hormone, metabotropic glutamate) but does not elaborate on their potential role in oyster reproduction. The paper correctly notes that these findings require further investigation, but a more detailed hypothesis on their potential function would be valuable.
The discussion lacks a dedicated limitations paragraph, which is a minor weakness. Key limitations include the small sample size (n=3 per group), the low transcriptome completeness (58.3% BUSCO score) which may have led to missed genes, and the correlational nature of the findings due to a lack of functional validation. Acknowledging these points would make the research more transparent and provide a clearer direction for future studies. The authors can add a subsection in the "Discussion" section titled "Limitations and Future Directions". This section should clearly state the issues of transcriptome completeness, small sample size, and the lack of functional validation. By doing so, you can provide a more balanced view of your work and outline a clear path for future research to build upon their findings. For example, you could suggest future studies that use techniques like RNA interference (RNAi) to validate the causal roles of genes like SOX9 or LASGLVamide in spawning regulation.
Conclusion:
The conclusion section at the end of the paper is quite broad. It could benefit from a more direct summary of the most significant findings.
Author Response
Comment: Title – The current title is effective and acceptable, but it can be improved to frame the study's contribution more impactfully. This can be done by highlighting the methods used (e.g., Transcriptomic and Histological Analysis) or by emphasizing the core finding (e.g., the changes in gene expression and the temporal nature of the research).
Response: Thanks for your suggestion. We have updated the title to: “Comparative transcriptomics provides insight into the neuroendocrine regulation of spawning in the black-lip rock oyster (Saccostrea echinata)”
Comment: Abstract – The abstract effectively summarizes the paper's key findings but can be improved for clarity. The main suggestion is to start with a more direct statement of the research problem—the challenges in managing black-lip rock oyster spawning—before detailing the study's purpose. The abstract should also condense the list of specific genes into broader categories to improve flow, while still highlighting key findings like the upregulation of the LASGLVamide gene in pre-spawn females and stress-response genes in post-spawn oysters. Finally, the conclusion could be rephrased to be more impactful by directly stating that the findings provide a molecular foundation for improving aquaculture and breeding strategies. These revisions would make the abstract easier for readers to quickly grasp the study's core contributions.
Response: Thanks for your suggestion. We have updated the Abstract based on your comments.
Comment: Introduction (Line 43–45) – "To address these issues, the NSW oyster industry is actively advancing through selective breeding for both species [3-6]". The authors could briefly explain what "advancing through selective breeding" entails and what traits are being selected for (e.g., disease resistance, faster growth rates). This would give the reader a better understanding of the applied context.
Response: We thank the reviewer for this helpful suggestion. We explained and updated it in the manuscript which is in the ”Introduction” section.
Comment: Introduction (Line 48–50) – "The black-lip rock oyster (Saccostrea echinata) is a prominent aquaculture species that has recently drawn researchers' attention as a new aquaculture commodity in Australia [9]". The introduction would benefit from a more explicit statement on why this species is of particular interest. What specific characteristics make it a "prominent aquaculture species"? Mentioning traits like fast growth, good taste, or hardiness would add valuable context.
Response: Thanks for pointing this out. We have now added this to the ‘Introduction’ section of the manuscript.
Comment: Introduction (Line 75–77) – "It is the visceral ganglia (VG), located in close association with the posterior adductor muscle, that controls the visceral mass, including the gonad [24]". The grammatical structure is not correct. "That controls" makes it unclear whether the muscle or the ganglia controls the visceral mass. Please revise.
Response: We appreciate the reviewer’s careful observation. The sentence has been revised to clarify that it is the visceral ganglia, and not the posterior adductor muscle, that controls the visceral mass. The revised text now reads: “The visceral ganglia (VG), located in close association with the posterior adductor muscle and responsible for controlling the visceral mass, including the gonad” which is in “Introduction” section and page number 02.
Comment: Introduction (Line 84–87) – "In this study, to better understand ... and reproduction-associated genes". This is a long, run-on sentence. It combines the overall objective with the specific methods. Try to split it into two sentences for better flow and clarity. Also, while the objectives are present in this paragraph, they are embedded in prose. Try to reformat the final paragraph to clearly state the study's aims.
Response: We thank the reviewer for this helpful suggestion. We have revised the text to separate the overall objective from the methodological detail and to present the study aims more explicitly. The revised text now reads. “In this study, we aimed to better understand the neuroendocrine regulation of spawning in S. echinata. To obtain this, we established a reference transcriptome derived from VG and gonads, which was then used in targeted identification of oyster neuropeptide and reproduction-associated genes” which is in “Introduction section”.
Comment: Introduction – General comment: The flow from the discussion of aquaculture challenges to the specific focus on neuroendocrine regulation could be smoother. Try to improve the transition between paragraphs after discussing the challenges of mass mortality and selective breeding to directly link the problem to the research question.
Response: Thanks for pointing this out. Based on this comment, we have updated the “Introduction” section of the manuscript.
Comment: Results – The transcriptome assembly statistics are presented in Table 1, including N50 length and BUSCO completeness. While these numbers are provided, a brief discussion of what these metrics mean in the context of the study would be beneficial. For instance, what does a BUSCO completeness of 58.3% imply for the study's ability to find all relevant genes? Please add a sentence or two to the beginning of the results section that interprets the assembly statistics.
Response: We thank the reviewer for this suggestion. In the revised discussion, we have expanded on the BUSCO ,which is “The assembly achieved an N50 length of 781 bp, and a BUSCO completeness of 58.3% which is within the range reported for other molluscan tissue-specific transcriptomics [27, 28]”, which is “Discussion” section.
Comment: Results – Figure 1 shows the histological sections, but the description is brief. In the caption for Figure 1, or in the text describing the figure, please provide a more detailed description of the key histological features for each stage.
Response: We thank the reviewer for this suggestion. We updated the manuscript which is in the “Result” section under the subheading of “Gonad histology at pre- and post-spawn S. echinata”.
Comment: Discussion – The discussion mentions the differential expression of several receptors (e.g., thyrotropin-releasing hormone, metabotropic glutamate) but does not elaborate on their potential role in oyster reproduction. The paper correctly notes that these findings require further investigation, but a more detailed hypothesis on their potential function would be valuable.
Response: We thank the reviewer for raising this important point. To the best of our knowledge, there are currently no published studies directly linking metabotropic glutamate receptors (mGluRs) with reproduction in oysters. However, mGluRs are known to mediate neuroendocrine signalling in vertebrates, where they influence hypothalamic–pituitary–gonadal axis activity and gametogenesis. We therefore consider this an intriguing area for future research, although direct evidence in oysters is not yet available.”
Comment: Discussion – The discussion lacks a dedicated limitations paragraph, which is a minor weakness. Key limitations include the small sample size (n=3 per group), the low transcriptome completeness (58.3% BUSCO score) which may have led to missed genes, and the correlational nature of the findings due to a lack of functional validation. Acknowledging these points would make the research more transparent and provide a clearer direction for future studies. The authors can add a subsection in the "Discussion" section titled "Limitations and Future Directions". This section should clearly state the issues of transcriptome completeness, small sample size, and the lack of functional validation. By doing so, you can provide a more balanced view of your work and outline a clear path for future research to build upon their findings. For example, you could suggest future studies that use techniques like RNA interference (RNAi) to validate the causal roles of genes like SOX9 or LASGLVamide in spawning regulation.
Response: We thank the reviewer for this suggestion, and we agree that acknowledging study limitations is important. In the current version of the Discussion, we had already addressed the main limitations raised (BUSCO, functional investigation of the neuropeptide genes) in the relevant sections where they directly relate to the interpretation of our findings. We believe that integrating these points contextually provides clarity without redundancy. However, we have carefully revised the text to ensure these limitations are highlighted more explicitly and clearly linked to future research directions. Due to the unavailability or limitations of getting enough broodstock of S. echinata from the farm (as a ripe bloodstock was very demanding and lucrative), we think the sample size (n=3) was sufficient for this study.
Comment: Conclusion – The conclusion section at the end of the paper is quite broad. It could benefit from a more direct summary of the most significant findings.
Response: We thank the reviewer for this suggestion. We have applied the comment on the “Conclusion” section.
Round 2
Reviewer 1 Report
Comments and Suggestions for Authors
The authors have satisfactorily responded to the comments made and have resolved the suggestions correctly, so it is considered that the manuscript is ready to be accepted for publication.
Reviewer 2 Report
Comments and Suggestions for Authors
I would like to thank authors for their effort.